# COVID-19-Related Burnout and Intention of Fully Vaccinated Individuals to Get a Booster Dose: The Mediating Role of Resilience

**DOI:** 10.3390/vaccines11010062

**Published:** 2022-12-27

**Authors:** Petros Galanis, Aglaia Katsiroumpa, Panayota Sourtzi, Olga Siskou, Olympia Konstantakopoulou, Theodoros Katsoulas, Daphne Kaitelidou

**Affiliations:** 1Clinical Epidemiology Laboratory, Faculty of Nursing, National and Kapodistrian University of Athens, 11527 Athens, Greece; 2Faculty of Nursing, National and Kapodistrian University of Athens, 11527 Athens, Greece; 3Department of Tourism Studies, University of Piraeus, 18534 Piraeus, Greece; 4Center for Health Services Management and Evaluation, Faculty of Nursing, National and Kapodistrian University of Athens, 11527 Athens, Greece

**Keywords:** COVID-19, burnout, intention, vaccination, resilience

## Abstract

Because an annual COVID-19 booster vaccine appears to be required to control the pandemic, identifying the factors that influence individuals’ decision to receive a booster dose is critical. Thus, our goal was to quantify the influence of COVID-19-related burnout on vaccination intention and to investigate the role of resilience in mediating the link between burnout and intention. We conducted a cross-sectional study with a convenience sample during October 2022. We used the COVID-19 burnout scale and the Brief Resilience Scale. The study sample included 1256 people who had received their primary COVID-19 vaccination. Among the participants, 34.1% reported being very likely to be vaccinated with a booster dose. COVID-19-related burnout was found to be inversely connected with vaccination intention. Moreover, our results suggested that resilience not only had a positive direct effect on vaccination intention but also mediated the relationship between burnout and vaccination intention. Although our study had limitations, such as a convenience sample and information bias, we demonstrate the critical role of resilience in reducing the impact of COVID-19-related burnout on the vaccination intention. Policymakers should develop and implement initiatives to address the issues of COVID-19-related burnout and enhance booster adoption by strengthening psychosocial resources such as resilience.

## 1. Introduction

As of 21 November 2022, more than 13 billion COVID-19 vaccine doses have been administered worldwide and 68.2% of the world population has received at least one dose of a vaccine [1]. The COVID-19 vaccines are vital to protect the public against the disease by reducing infection rate, severity, hospitalization, and mortality [2,3]. For instance, estimations show that vaccines prevent almost 20 million deaths from COVID-19 worldwide in one year [4]. However, the vaccination rate in low-income countries is very low as only 23.6% of citizens in these countries have received at least one dose of a COVID-19 vaccine [1]. Moreover, the effectiveness of primary vaccination (a) against severe disease decreases within six months [5,6] and (b) against new variants (e.g., the Delta and Omicron variants) is limited [7,8]. Therefore, several countries have already approved booster doses, especially in high-risk groups. Booster vaccination improves the neutralization of serum antibodies against the new Omicron variant but produces incomplete immunity [9]. Thus, future vaccination against COVID-19 seems to be necessary with pharmaceutical companies trying to develop new generation COVID-19 vaccines that provide better and broader protection.

### 1.1. Intention to Vaccinate

Because COVID-19 vaccination is a cost-effective [10,11] and safe public health intervention [12,13], efforts should be taken to improve the willingness of individuals to accept booster doses. Public acceptance of COVID-19 vaccines and booster doses is influenced by several factors, such as gender, age, educational level, income, insurance, racial/ethnic minority, safety, efficacy, effectiveness and side effects of vaccines, previous flu vaccination, trust, information sufficiency, conspiracy beliefs, social influence, political roles, fear, and anxiety [14,15,16,17]. Future vaccination with booster doses or new vaccines seems to be the best defense mechanism society has against COVID-19. Policy makers should clarify the effectiveness, safety, and side effects of booster doses in order to foster vaccine confidence and decrease vaccine hesitancy [16].

### 1.2. Burnout

Since the World Health Organization has already recognized vaccine hesitancy as one of the ten biggest threats to global health, the COVID-19 vaccine hesitancy of people to accept future vaccinations could cause a significant problem in controlling the pandemic worldwide [18]. Moreover, the “vaccine fatigue” phenomenon [19] may compromise individuals’ vaccination intention against COVID-19. Several factors increase vaccine fatigue such as vaccine side effects, an increased number of doses, a lack of trust in the government and the media, and misinformation about the need for vaccination [19]. COVID-19 vaccine fatigue is probable because, almost three years after the onset of the pandemic, new variants continue to cause new pandemic waves and rapidly escalate COVID-19 cases [20]. Thus, people’s trust in COVID-19 vaccines may be compromised. In addition, conspiracy theories and misinformation around COVID-19 vaccines induce a decline in intent to vaccinate [21,22]. Almost three years into the pandemic, it is overwhelming to still keep up with the measures against COVID-19 [23]. Moreover, the pandemic causes a tremendous effect on people’s mental health, thereby increasing fear, depression, anxiety, stress, insomnia, and post-traumatic stress disorder [24]. For instance, a meta-analysis found that the prevalence of depression, anxiety, insomnia, and post-traumatic stress disorder among populations affected by the COVID-19 pandemic was 16%, 15.2%, 23.9%, and 22%, respectively [25].

Although the literature on the mental health problems caused by the pandemic in the general population is extensive, only two studies have investigated COVID-19-related burnout in the general population [26,27]. In particular, Yildirim & Solmaz (2022) [27] found that COVID-19 stress predicts COVID-19 burnout through resilience while Lau et al. (2022) [26] found a positive correlation between burnout and fear. Research on burnout during the COVID-19 pandemic has been mainly conducted on healthcare workers [28,29,30]. To the best of our knowledge, no study until now has investigated the impact of COVID-19-related burnout on individuals’ intention to get vaccinated against COVID-19. Therefore, our first hypothesis was as follows:

**H1.** 
*COVID-19-related burnout would have a direct effect on booster vaccination intention. Specifically, we hypothesized that the higher the levels of burnout, the lower the booster vaccination intention.*


### 1.3. Resilience

Resilience is commonly defined as the ability to cope with difficult or challenging life situations, e.g., trauma, stress, tragedy, threats, etc. [31]. In addition, resilience refers to the ability to recover quickly from setbacks and difficult situations [32]. During the pandemic, Kimhi et al. (2020) [33] found a negative correlation between resilience and distress symptoms in the general population while Jose & Dhandapani (2020) [34] found a negative correlation between resilience and burnout among frontline nurses. Moreover, Yildirim et al. (2022) [27] found that resilience was a mediator in the relationship between COVID-19 stress and COVID-19 burnout in the general population while Mo et al. (2022) [35] conducted a mediation analysis in a sample of healthcare workers and discovered that resilience improved the COVID-19 vaccination intention of healthcare workers both directly and indirectly through life satisfaction and levels of stigma. Thus, based on the current literature, we investigated the following second hypothesis:

**H2.** 
*Resilience would be a mediator in the relationship between COVID-19-related burnout and vaccination intention.*


In short, the aim of our study was to estimate the direct effect of COVID-19-related burnout on vaccination intention and to analyze the mediating effect of resilience on the relationship between burnout and intention (Figure 1).

## 2. Materials and Methods

### 2.1. Study Design and Participants

We conducted a cross-sectional study in Greece during October 2022. Because the questionnaire was in Greek, adults who understood the language could take part in our study. Furthermore, because our goal was to estimate individuals’ willingness to receive a booster dosage against COVID-19, we included in our study those who had completed the initial immunization against COVID-19. We created an anonymous version of the questionnaire using Google forms and then distributed it over social media networks and email connections. In particular, we posted the questionnaire in public groups and our personal social media pages. We posted the questionnaire again as a reminder a week after the first post. In order to avoid selection bias, we did not post the questionnaire in public groups categorized by specific profession, gender, etc. In addition, we shared the web link via email to our personal contacts. Moreover, we encouraged participants to invite others to complete the study questionnaire by forwarding the web link and post the web link on their social media page. Thus, we employed a snowball sampling technique in order to obtain a convenience sample. We informed the participants about the aim of the study before the first question, and they gave their informed consent to participate. Therefore, participation in the study was anonymous and voluntary. We applied the principles of the Declaration of Helsinki. The Ethics Committee of the Department of Nursing, National and Kapodistrian University of Athens, approved the study protocol (reference number; 370, 02-09-2021).

To estimate the required sample size for our study, we used the Hair et al. (2017) rule of thumb, which states that participants in a mediation analysis should be at least 10 times the number of study variables [36]. Thus, the required sample size for our study was 280 participants (=28 variables * 10 = 280). Moreover, considering that the reference population of fully vaccinated individuals in Greece was 7.64 million at the time of the study, the minimum sample size would be 666 participants with a confidence level of 99%, a margin of error of 5%, and a population proportion of 50%. We increased our sample size in order to decrease the random error of our measurements.

### 2.2. Measures

#### 2.2.1. Demographic Data

We asked participants to report their gender (females or males), age (continuous variable), highest educational level (elementary school, high school, university degree, MSc/PhD diploma), chronic condition (no or yes), self-assessment of health status (very poor, poor, moderate, good, very good), COVID-19 infection (no or yes), booster doses (no or yes), and side effects because of COVID-19 vaccination (a scale from 0 [none] to 10 [many]).

#### 2.2.2. Intention to Vaccinate

We used the following question to assess individuals’ willingness to get vaccinated against COVID-19 with a booster dose: “The Health Ministry suggests a booster dose against COVID-19 for the fully vaccinated individuals with primary doses. Given that this vaccination will not be mandatory, how likely do you believe you will be to acquire a booster dose?”. Answers were on an eleven-point scale from 0 (extremely unlikely) to 10 (extremely likely). We used a priori cut-off points to the 0–10 scale in order to categorize participants in terms of their vaccination intention; scores ≤ 2 as “very unlikely to be vaccinated with a booster dose”, scores of three to seven as “uncertain”, and scores ≥ 8 as “very likely”.

#### 2.2.3. Burnout

We used the COVID-19 burnout scale (COVID-19-BS) to measure COVID-19-related burnout [37]. The COVID-19-BS is a valid instrument in Greek, and it was developed to specifically measure COVID-19-related burnout in the general population. The COVID-19-BS consists of 13 items (e.g., “I feel tired of getting vaccinated against coronavirus”) with answers in a Likert scale from 1 (strongly disagree) to 5 (strongly agree). The COVID-19-BS covers three dimensions of burnout: emotional, physical, and burnout due to measures against COVID-19. The overall score on the COVID-19-BS ranges from 1 (low level of burnout) to 5 (high level of burnout).

#### 2.2.4. Resilience

We used the Brief Resilience Scale (BRS) to measure participants’ resilience [38]. The BRS consists of six items (e.g., “I tend to bounce back quickly after hard times”) and answers are in a five-point Likert scale from 1 (strongly disagree) to 5 (strongly agree). The total BRS score ranges from 1 (low resilience) to 5 (high resilience). The BRS has been validated in Greek [39]. Cronbach’s alpha for the BRS was 0.82 in our study.

### 2.3. Statistical Analysis

Descriptive statistics (i.e., absolute frequency, percentage, mean, standard deviation [SD]) were performed on demographic variables to summarize gender, age, educational level, chronic condition, self-assessment of health status, COVID-19 infection, booster doses, and side effects because of COVID-19 vaccination. Scores on the vaccination intention scale, COVID-19-BS, and BRS followed normal distribution. Thus, we estimated mean and standard deviation for these three scales. We also estimated the correlation between the scales using Pearson’s correlation coefficient.

Since the COVID-19 burnout scale is a newly developed tool, we performed a validity and reliability analysis to check the psychometric properties of the tool in our population. Validity analysis included face validity and confirmatory factor analysis. We performed interviews with 10 individuals to check the face validity. We conducted a confirmatory factor analysis using AMOS (version 23) to test the structure of the COVID-19-BS. The literature suggests the following cut-off values as good fit indices: chi-square divided by degree of freedom (x^2^/df) below 3; a root mean square error of approximation (RMSEA) below 0.08; and a goodness-of-fit index (GFI), adjusted goodness-of-fit index (AGFI), Tucker–Lewis index (TLI), incremental fit index (IFI), normed fit index (NFI), and comparative fit index (CFI) above 0.95 [40,41,42,43]. We calculated Cronbach’s alpha coefficient, Guttman’s split-half coefficient, and the corrected item–total correlations to assess the reliability of the COVID-19-BS. Acceptable values for Cronbach’s alpha coefficient and Guttman’s split-half coefficient are above 0.7 [44], while for the corrected item–total correlation, the acceptable value is above 0.3 [45].

To test our predicted mediation model, we employed the PROCESS macro (model 4) [46]. We reported regression coefficients (β) and squared multiple correlations (R^2^) with conventional effect sizes: 0.01–0.059: small; 0.06–0.139: moderate; ≥0.14: large [47]. We conducted a bootstrapping procedure with 5000 re-samples in order to estimate indirect effect and 95% confidence intervals (95% CI) [48]. The mediating effect was deemed statistically significant if zero was not included in the 95% CI. In addition, we calculated the Sobel test [49] to assess whether the indirect effect of the COVID-19-related burnout on the vaccination intention through resilience was significant. Then, we employed a confirmatory factor analysis to check the robustness of the mediation analysis results. We used the AMOS software to perform the confirmatory factor analysis and assess the fit of model. We utilized several standard criteria to evaluate the goodness-of-fit model. In particular, x^2^/df below 5, RMSEA below 0.08, and TLI, IFI, and CFI above 0.9 indicate a good fit of model [40,41,42,43]. Moreover, we calculated standardized estimate effects (total, indirect, and direct effects) and we examined the mediating effect of resilience with a bias-corrected bootstrap 95% CI. We based our estimates on these 5000 bootstrap samples. *p*-values less than 0.05 were considered as statistically significant. We used IBM SPSS 21.0 (IBM Corp. Released 2012. IBM SPSS Statistics for Windows, Version 21.0. Armonk, NY, USA: IBM Corp.) for statistical analysis.

## 3. Results

### 3.1. Study Sample

A total of 1256 individuals completed the online survey. Characteristics of the sample are shown in Table 1. The mean age of participants was 39.2 years, with a range from 19 to 80. The study sample comprised 69.9% females while 19.4% of the participants had a chronic condition. In terms of health status, 2.7% reported a very poor/poor level of health status, 7.6% a moderate level, and 89.7% a good/very good level. Among the participants, 69.4% had been infected with the SARS-CoV-2, 84.2% had had a booster dose, and 78.2% had experienced side effects because of past COVID-19 vaccination.

### 3.2. Associations among Study Variables

We presented means, standard deviations, medians, and correlations of the study variables in Table 2. Our results indicated a mean of 4.785 (SD = 3.592) for COVID-19 vaccination intention, 3.028 (SD = 1.090) for COVID-19-related burnout, and 3.419 (SD = 0.733) for resilience. Among the participants, 31.8% (n = 400) said they were extremely unlikely to get vaccinated with a booster dose, 34.1% (n = 428) said they were unsure, and 34.1% (n = 428) said they were very likely to be vaccinated. COVID-19-associated burnout was found to be inversely related to vaccination intention and resilience. In addition, resilience was found to be positively connected to vaccination intention.

### 3.3. Validity and Reliability of the COVID-19 Burnout Scale

Face validity of the COVID-19-BS was excellent as we did not make any changes on the 13 items of the tool after the interviews with 10 individuals. All participants considered the 13 items as clear and comprehensive. Then, we performed a confirmatory factor analysis to test the three-factor structure of the COVID-19-BS. The eight goodness-of-fit indices indicated a very good fit of model and confirmed the structure of the COVID-19-BS in our study: x^2^/df = 2.368, RMESA = 0.033, GFI = 0.987, AGFI = 0.974, TLI = 0.990, IFI = 0.994, NFI = 0.990, and CFI = 0.994. In addition, correlations between the three factors ranged from 0.52 to 0.87 and were statistically significant (Appendix A). Moreover, the standardized regression weights between the items and the factors ranged from 0.595 to 0.908 (*p* < 0.001 in all cases).

In addition, the reliability of the COVID-19-BS in our study was excellent. In particular, Cronbach’s alpha for the COVID-19-BS was 0.921 and ranged from 0.862 to 0.900 for the three scales. In addition, Guttman’s split-half coefficient was 0.787, and the corrected item–total correlations for the 13 items ranged from 0.531 to 0.739 (Appendix A).

### 3.4. Mediation Analysis

We used the PROCESS macro in conjunction with model 4 to conduct a mediation study to investigate the indirect influence of COVID-19-related burnout on vaccination intention via resilience. Mediation analysis showed that COVID-19-related burnout influenced vaccination intention both directly and indirectly through the mediation effect of resilience (Figure 2, Table 3). In particular, COVID-19-related burnout was a significant predictor of resilience (β = −0.2491, *p* < 0.0001) and vaccination intention (β = −0.4292, *p* < 0.0001). COVID-19-related burnout explained 21.0% of the variance in vaccination intention. As a result, empirical evidence for Hypothesis 1 was established, namely, the higher the levels of burnout, the lower the vaccination intention. Moreover, COVID-19-related burnout had a significant indirect effect on vaccination intention through resilience (β = 0.0656, 95% CI = 0.0002 to 0.1410, and standard error = 0.0353). Resilience partially mediated the effect of COVID-19-related burnout on vaccination intention. Therefore, our data supported Hypothesis 2 that resilience was a mediator in the relationship between COVID-19-related burnout and vaccination intention. COVID-19-related burnout and resilience collectively accounted for 34.6% of the variance in vaccination intention. Moreover, the Sobel test confirmed the mediating effect of resilience on the relationship between burnout and vaccination intention (*p*-value = 0.040, Sobel test = 2.054, standard error = 0.032). Mediation model summary information is presented in Table 3.

### 3.5. Confirmatory Factor Analysis

The model-fitting results from the confirmatory factor analysis showed that all fitting indices were acceptable (Table 4). Thus, our constructed model fitted very well. The confirmatory factor analysis confirmed our two hypotheses. Regarding the first hypothesis, we found that COVID-19-related burnout was negatively associated with vaccination intention (standardized coefficient [β] = −0.421, *p* < 0.0001). The final model with standardized coefficients is shown in Figure 3. Moreover, the confirmatory factor analysis confirmed our second hypothesis, since we found that resilience partially mediated the relationship between COVID-19-related burnout and intention (standardized indirect effect = 0.044, 95% bias-corrected CI = 0.009 to 0.081, *p* = 0.011). Table 5 shows the detailed results of the mediating effects tested by the bias-corrected bootstrap method.

## 4. Discussion

We aimed to evaluate whether COVID-19 vaccination intention can be predicted directly or indirectly by COVID-19-related burnout with a potential mediation of resilience. To the best of our knowledge, this study is the first of its kind worldwide. Overall, our results support the two study hypotheses. In particular, COVID-19-related burnout was found to be inversely connected with vaccination intention. Moreover, we found that resilience partially mediated the relationship between COVID-19-related burnout and vaccination intention.

According to our study, 34.1% of the participants were likely to take a booster dose. This finding is alarming, since similar studies in the United Kingdom, USA, Australia, and Malaysia reported higher levels of booster intent at 73%, 76%, 67%, and 81.2%, respectively [50,51,52]. This disparity in vaccination intentions could be attributable to the timing of the research, as we collected our data in October 2022, whereas the other investigations were done between September 2021 and February 2022. It is probable that, as time passes, individuals adopt effective coping strategies to deal with the pandemic, experience less fear against COVID-19, and feel safer. Therefore, as people become more familiar with the pandemic, their willingness to get a booster dose is reduced.

We discovered that COVID-19-related burnout reduced vaccine intention, as predicted. Although there is no available evidence regarding the association between COVID-19-related burnout and vaccination intention, it seems reasonable for there to be a negative association between these two variables. COVID-19 vaccine fatigue is quite probable almost two years after the COVID-19 vaccine rollouts and several doses. Moreover, a recent systematic review shows that COVID-19 conspiracy beliefs continue to have negative consequences affecting vaccination intentions, protective behaviors, and psychological well-being [53]. Although boosted individuals exhibit fewer COVID-19 conspiracy beliefs compared with individuals with primary vaccination [54], we should continue to support them in order to maintain their trust of COVID-19 vaccinations. Booster doses are necessary to maintain the immune response of individuals against the SARS-CoV-2; however, there are limited data regarding the effectiveness, safety, and side effects of booster doses. In addition, experiences of side effects due to previous COVID-19 vaccines doses may compromise the trust of the general public, even though side effects are usually mild and self-limited while severe reactions are rare [55]. Furthermore, as expected, infections, hospitalizations and deaths have also occurred among fully vaccinated individuals. Booster vaccination dose recipients, on the other hand, have better protection against illness, hospitalization, and mortality [56]. Thus, COVID-19-related exhaustion, COVID-19 conspiracy theories, fake news, and confusion as a result of an overload of constant information could all work together to diminish vaccination intentions [57]. Since an annual COVID-19 booster vaccine seems to be necessary, especially for vulnerable groups, policy makers should make clear that COVID-19 booster doses are safe and effective by providing valid and updated surveillance data in order to decrease vaccine hesitancy [16].

The most interesting finding of our study was to demonstrate that resilience not only had a direct effect on vaccination intention but also mediated the relationship between COVID-19-related burnout and vaccination intention. This suggests that COVID-19-related burnout could directly or indirectly, through lessening resilience, decrease vaccination intention. In other words, individuals with higher levels of COVID-19-related burnout had a tendency to experience less resilience which in turn contributed to a decrease in their vaccination intention. The literature suggests the positive role of resilience in the current pandemic. In particular, resilience improves COVID-19 vaccination intention among healthcare workers both directly and indirectly through levels of stigma and life satisfaction [35]. In addition, resilience is a mediator on the relationship between meaningful living and psychological health among young adults [58]. Moreover, resilience partially mediates the relationship between COVID-19 stress and burnout in the general population [27]. Several other studies during the pandemic era suggest the positive impact of resilience as a mediator, since it has a buffering effect on the development of burnout, depression, post-traumatic stress disorder, and anxiety [59,60,61,62,63]. In addition, evidence suggests that resilience compensates for the negative influence of loneliness, social isolation, and stress on psychological well-being and life satisfaction in the context of a pandemic [64,65,66]. These results suggest that character strengths like resilience help individuals to adopt appropriate coping strategies and adapt to difficult situations. Thus, personal psychological resources such as resilience are important aspects of meaning-focused preventions and could promote positive attitudes during the pandemic such as COVID-19 vaccination uptake.

### Limitations

We acknowledge that our study is not without limitations. First, our sample cannot be considered representative of the general Greek population, despite the large sample size, since we used a non-representative sampling method. For example, participants mainly represented females, young adults, and highly educated individuals. In our study, males, the elderly, and lower educated individuals were underrepresented. Since a random sample from the general public could not be achieved, we have to dispense with demographic inclusion criteria. Second, we used self-reported questionnaires to measure burnout, resilience, and vaccination intention. Therefore, an information bias is probable in our study, and our results may differ from actual behavior. Third, since we used cross-sectional data, only correlation evidence can be obtained from our mediation analysis. Follow-up studies could add invaluable information to understand a causal relation from COVID-19-related burnout to vaccination intention through resilience. Fourth, we only investigated resilience as a possible mediator in the relationship between COVID-19-related burnout and vaccination intention while there may be other potential mediators in this relationship. Fifth, we collected our data during a specific time and individuals’ attitudes may change in the future. Therefore, further studies should be conducted in order to establish more robust results. Finally, it is always possible that unmeasured confounding factors affect the relationship between COVID-19-related burnout and vaccination intention. For example, future research could investigate the possible role of climate change, natural resource depletion, and wars as confounders [67].

## 5. Conclusions

Our findings suggest a negative relationship between COVID-19-related burnout and vaccination intention while resilience partially mediates this relationship. Thus, our study demonstrates the critical role of resilience in reducing the impact of COVID-19-related burnout on the vaccination intention. For that reason, policy makers should implement interventions to improve booster vaccination by building resilience, decreasing COVID-19-related burnout, supporting appropriate coping strategies, and debunking fake news and misinformation. Although our study had several limitations, such as a convenience sample and information bias, our results can be used by policy makers to develop and implement measures to deal with the challenges of COVID-19-related burnout and to improve booster uptake.

## Figures and Tables

**Figure 1 vaccines-11-00062-f001:**
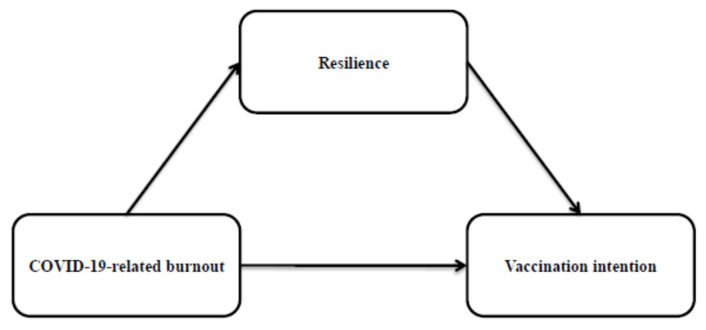
Structural model depicting the relationships between COVID-19-related burnout, resilience, and vaccination intention.

**Figure 2 vaccines-11-00062-f002:**
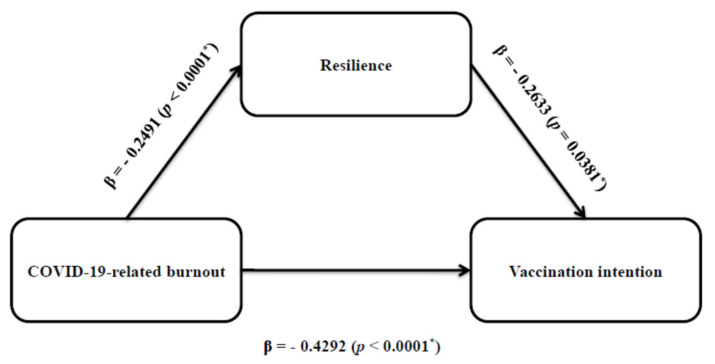
Structural mediation model using PROCESS macro (model 4) with path coefficients (β) and *p*-values of resilience as the mediator in the relationship between COVID-19-related burnout and vaccination intention. * Statistically significant.

**Figure 3 vaccines-11-00062-f003:**
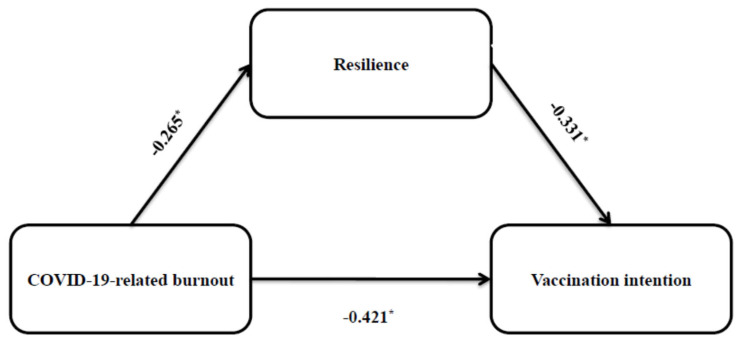
Standardized coefficients (β) from confirmatory factor analysis on the relationship between COVID-19-related burnout, vaccination intention, and resilience. * Statistically significant.

**Table 1 vaccines-11-00062-t001:** Characteristics of the sample (N = 1256).

Variable	N	%
Gender		
Females	878	69.9
Males	378	30.1
Age, mean, standard deviation	39.2	11.9
Highest educational level		
Elementary school	12	1.0
High school	212	16.9
University degree	542	43.1
MSc/PhD diploma	490	39.0
Chronic condition		
No	1012	80.6
Yes	244	19.4
Self-assessment of health status		
Very poor	28	2.2
Poor	6	0.5
Moderate	96	7.6
Good	678	54.0
Very good	448	35.7
COVID-19 infection		
No	384	30.6
Yes	878	69.4
Booster doses		
No	198	15.8
Yes	1058	84.2
Side effects because of COVID-19 vaccination, mean, standard deviation	2.5	2.4

**Table 2 vaccines-11-00062-t002:** Descriptive statistics of the study variables and correlation matrix between them.

Variable	Mean	SD	Median	1.	2.	3.
1. COVID-19 vaccination intention	4.785	3.592	5.000	-	−0.186 **	0.057 *
2. COVID-19-related burnout	3.028	1.090	3.077		-	−0.407 **
3. Resilience	3.419	0.733	3.500			-

SD—standard deviation. * *p* < 0.05 and ** *p* < 0.001.

**Table 3 vaccines-11-00062-t003:** Mediation model summary information for the COVID-19-related burnout using the model depicted in Figure 2.

	Outcome											
	M (Resilience)						Y (COVID-19 Vaccination Intention)					
					95%						95%	
Antecedent	Coeff.	SE	t	*p*	LLCI	ULCI	Coeff.	SE	t	*p*	LLCI	ULCI
X (COVID-19-related burnout)	−0.2491	0.0180	−13.84	<0.0001 *	−0.2844	−0.2138	−0.4292	0.0864	−4.97	<0.0001 *	−0.5988	−0.2597
M (resilience)	-	-	-	-	-	-	−0.2633	0.1268	−2.08	0.0381 *	−0.5121	−0.0145
Constant	3.6190	0.1523	23.76	<0.0001 *	3.2201	3.9178	4.0585	0.8209	4.95	<0.0001 *	2.4479	5.6691
	R^2^ = 0.2100, F = 30.01, *p* < 0.001 *						R^2^ = 0.3461, F = 54.71, *p* < 0.001 *					

Coeff—regression coefficient; LLCI—lower limit of confidence interval; M—mediator variable; R^2^—squared multiple correlations; SE—standard error; ULCI—upper limit of confidence interval; X—independent variable; and Y—dependent variable. * Statistically significant.

**Table 4 vaccines-11-00062-t004:** Goodness-of-fit statistics for confirmatory factor analysis model.

	X^2^	df	x^2^/df	RMSEA	TLI	IFI	CFI
Our model	4.725	27	1.545	0.021	0.971	0.992	0.991
Acceptable values			<5	<0.08	>0.90	>0.90	>0.90

**Table 5 vaccines-11-00062-t005:** Standardized total, indirect, and direct effects from confirmatory factor analysis on the mediating role of resilience on the association between COVID-19-related burnout and vaccination intention.

Variables	Standardized Estimates	Bootstrap		
		Bias-Corrected 95% CI		*p*-Value
		Lower Bounds	Upper Bounds	
COVID-19-related burnout → vaccination intention (total effects)	−0.160	−0.239	−0.088	<0.0001
COVID-19-related burnout → vaccination intention (indirect effects)	0.044	0.009	0.081	0.011
COVID-19-related burnout → vaccination intention (direct effects)	−0.204	−0.286	−0.127	<0.0001

## Data Availability

The data presented in this study are available on request from the corresponding author.

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
