# Peer review of "COVID-19-Related Burnout and Intention of Fully Vaccinated Individuals to Get a Booster Dose: The Mediating Role of Resilience"

_vaccines, 2022, doi:10.3390/vaccines11010062_

Round 1
Reviewer 1 Report
I would like to thank the editors of "Vaccines" for their review of this paper.
The authors make a good presentation of their working hypothesis and objectives, using two appropriate tools, namely COVID burnout and resilience.
The statistical analysis is correct and the results are consistent. The limitations presented by the authors should be taken into account when extrapolating the results.
This is an interesting topic, as it is expected that the population will have to be revaccinated after 2 years of the pandemic, and it is interesting to know your opinion on the vaccine.
This is a cross-sectional observational study, with purposive sampling. The sample should be extended to other under-represented populations such as males with low levels of education. But this can no longer be solved, due to the type of study and sampling. It would force a redo of the work.
The references are appropriate.
Tables and figures are appropriate.
The work does not require changes in its current state.
Author Response
Dear Reviewer, we are grateful for your positive comments.
Reviewer 2 Report
This paper aimed to estimate the effect of COVID-19-related burnout on vaccination intention and to analyze the mediating role of resilience on the relationship between burnout and intention. The COVID-19 burnout scale was used to measure COVID-19-related burnout and the Brief Resilience Scale to measure resilience. The participants were 1256 individuals that have completed the primary vaccination against COVID-19. The results suggested that resilience not only had a positive direct effect on vaccination intention but also mediated the relationship between burnout and vaccination intention. The implications for policymakers are discussed.
This is an interesting paper that provides empirical evidence about the role of resilience in COVID-19-related burnout. The sample size is very large so the concerns about the opportunity sampling are ameliorated. However the limitations of the study are properly reported, and the methodology is clearly presented.
The paper is worth publishing,
However, for reinforcing the empirical findings I would ask to add the following:
1. Please provide Confirmatory Factor analysis for the measurements (the unidimensional constructs) with the instrument used.
2) I would suggest performing a Sobel test for the mediation analysis.
These can easily be carried out-the revision is very minor.
Author Response
Dear Reviewer,
Thank you for giving us the opportunity to revise our manuscript entitled "COVID-19-related burnout and intention of fully vaccinated individuals to get a booster dose: The mediating role of resilience". We would also like to thank you for your insightful comments and suggestions on how to improve our manuscript. We respectfully tried to address the issues raised and to revise our manuscript accordingly. We hope that our revision will reach the high standards of the journal “Vaccines”.
We are grateful for your comments. You really help us to improve our manuscript. We apply all your suggestions in our manuscript.
Please see the attached file for details and the revised manuscript.
Also, we made changes in the manuscript according to the other Reviewers’ instructions.
We look forward to hearing from you
Best Regards
The authors

Reviewer 3 Report
REVIEWER'S REPORT
Manucsript title: COVID-19-related burnout and intention of fully vaccinated 2 individuals to get a booster dose: The mediating role of resilience. (Authors: Petros Galanis, Aglaia Katsiroumpa 1, Panayota Sourtzi , Olga Siskou, Olympia Konstantakopoulou, Theodoros Katsoulas, and Daphne Kaitelidou.
The purpose of this manuscript was to quantify the impact of COVID-19-related fatigue on vaccination intention and to evaluate the role of resilience in modulating the burnout-intention relationship. COVID-19-associated burnout was discovered to be inversely related to vaccination intent. According to the findings, resilience not only had a direct favorable influence on vaccination intention, but it also moderated the association between COVID-19-related burnout and vaccination intention. Finally, the study highlighted the importance of resilience in mitigating the influence of COVID-19-related fatigue on vaccination intention.
This manuscript could be accepted with some editing. I recommend proofreading the whole text of the manusript as it may have a signifcant impact on the quality of the work.
Namely:
In abstract (page 1). In lines 17-20, the text should be praphrased as follows "Because an annual COVID-19 booster vaccine appears to be required to control the pandemic, identifying the factors that influence individuals' decision to receive a booster dose is critical. Thus, our goal was to quantify the influence of COVID-19-related burnout on vaccination intention and to investigate the role of resilience in moderating the link between burnout and intention." In lines 23-26 might be phrased as " The study sample included 1256 people who had received their primary COVID-19 vaccination. COVID-19-related burnout was found to be inversely connected with vaccination intention." In lines 29-31, the sentence could be replaced by "Policymakers should develop and implement initiatives to address the issues of COVID-19-related burnout and enhance booster adoption by strengthening psychosocial resources such as resilience."
In Materials and Methods, Study design and participants (page 3). To my mind, the text should be paraphrased as "...Because the questionnaire was in Greek, adults who understood the language could take part in our study. Furthermore, because our goal was to estimate individuals' willingness to receive a booster dosage against COVID-19, we included in our study those who had completed the initial immunization against COVID-19. We created an anonymous version of the questionnaire using Google forms and then distributed it over social media networks and email connections...." In lines 128-130, could be written as "To estimate the required sample size for our study, we used Hair et al. (2017) rule of thumb, which states that participants in a mediation analysis should be at least 10 times the number of study variables [36]."
In page 4, in line 135, minor correction "We asked participants to report their gender..." Lines 141-144 should be paraphrased as "We used the following question to assess individuals' willingness to get vaccinated against COVID-19 with a booster dose: ... Given that this vaccination will not be mandatory, how likely do you believe you will be to acquire a booster dose?"
In Statistical analysis (page 4), the text in lines 171-172 should be rewritten as "We also estimated the correlation between the scales using Pearson's correlation coefficient. To test our predicted mediation model, we employed PROCESS macro (model 4) [40]."
In Results (page 6), in lines 197-201, the text should be paraphrased as "Among the participants, 31.8% (n=400) said they were extremely unlikely to get vaccinated with a booster dose, 34.1% (n=428) said they were unsure, and 34.1% (n=428) said they were very likely to be vaccinated.
COVID-19-associated burnout was found to be inversely related to vaccination intention and resilience.
In addition, resilience was found to be positively connected to vaccination intention."
In Mediation Analysis (page 6), the sentences in lines 208-210 should be paraphrased as "We used the PROCESS macro in conjunction with model 4 to conduct a mediation study to investigate the indirect influence of COVID-19-related burnout on vaccination intention via resilience." In lines 214-215, should be written as "As a result, empirical evidence for Hypothesis 1 was established, namely..."
In Discussion (page 6),the lines 240-241 should be written as "...COVID-19-related burnout was found to be inversely connected with vaccination intention." In lines 247-249, the text should be paraphrased as "This disparity in vaccination intentions could be attributable to the timing of the research, as we collected our data in October 2022, whereas the other investigations were done between September 2021 and February 2022." In line 253, the sentence should be written as "We discovered that COVID-19-related burnout reduced vaccine 253 intention, as predicted." In line 267, it should be written as "Furthermore, as expected..."; in lines 268-271, the sentences should corrected as " Booster vaccination dose recipients, on the other hand, have better protection against illness, hospitalization, and mortality [49]. Thus, COVID-19-related exhaustion, COVID-19 conspiracy theories, fake news, and confusion as a result of an overload of constant information could all work together to diminish vaccination intentions."
The limitations of this study, in my opinion, should be briefly mentioned in the Abstract and Conclusion.
Author Response
Dear Reviewer,
Thank you for giving us the opportunity to revise our manuscript entitled "COVID-19-related burnout and intention of fully vaccinated individuals to get a booster dose: The mediating role of resilience". We would also like to thank you for your insightful comments and suggestions on how to improve our manuscript. We respectfully tried to address the issues raised and to revise our manuscript accordingly. We hope that our revision will reach the high standards of the journal “Vaccines”.
We are grateful for your comments. You really help us to improve our manuscript. We apply all your suggestions in our manuscript.
We mention the limitations of our study in the Abstract and Conclusion.
Please see the revised manuscript.
Also, we made changes in the manuscript according to the other Reviewers’ instructions.
We look forward to hearing from you
Best Regards
The authors